# The Efficacy of Protective Nets Against *Drosophila suzukii*: The Effect of Temperature, Airflow, and Pest Morphology

**DOI:** 10.3390/insects16030253

**Published:** 2025-03-01

**Authors:** Antonio J. Álvarez, Rocío M. Oliva, Jaime Martínez-Valderrama

**Affiliations:** 1Departamento de Ingeniería, Universidad de Almeria, 04120 Almeria, Spain; 2Departamento Agroforestal y Ambiental, Facultad de Ciencias y Artes, Universidad Católica de Ávila, Calle Canteros S/N Ávila, 05005 Ávila, Spain; rocio.oliva@ucavila.es; 3Estación Experimental de Zonas Áridas, CSIC, Ctra. Sacramento s/n, La Cañada, 04120 Almería, Spain; jaimonides@eeza.csic.es

**Keywords:** spotted wing drosophila, crop protection, control method, textile physical barriers, insect-proof screens, netting barriers, agrotextiles

## Abstract

The spotted wing drosophila (*Drosophila suzukii*) is a harmful insect that damages many types of fruit, causing serious losses for farmers around the world. Unlike other fruit flies, this pest can lay its eggs inside healthy, ripening fruit, making control very difficult. Chemical treatments often have negative effects on the environment, so farmers are looking for safer and more sustainable methods. Using protective nets is a promising solution because they create a physical barrier that prevents insects from reaching crops. In this study, we tested nine different types of nets to see which ones work best against this pest. We also studied how wind speed, temperature, and the size of the flies affect net performance. We found that high temperatures and strong winds reduce the nets’ ability to block the insect. We also discovered that flies raised at lower temperatures are larger, making it easier for them to get trapped by the net. Some nets were completely effective, offering farmers a reliable way to protect their crops without chemicals. These results can help farmers choose the best nets and improve pest management, reducing damage to crops while promoting more environmentally friendly practices.

## 1. Introduction

The spotted wing drosophila (SWD), *Drosophila suzukii* Matsumura (Diptera: Drosophilidae), is an invasive pest native to Asia that has become a serious pest worldwide, affecting a wide range of fruit crops. Since 2008, this pest was simultaneously described both in Europe [1,2] and North America [3] and subsequently also in South America [4]. This polyphagous pest has a great impact on the production of blackberries, blueberries, raspberries, strawberries, cherries, grapes, peaches, and other thin-skinned and stone fruits all over the world [5,6]. The rapid spread of *D. suzukii* across countries has demonstrated this species’ ability to adapt to newly invaded areas as an r-strategist, resulting in considerable agricultural losses [7] and significant economic damage [1,4,8,9,10,11,12,13,14,15,16,17].

SWD’s preference for laying eggs in ripening and ripe small, soft, and stone fruits still attached to the plant makes this pest a severe and harmful threat to fruit production. What makes female *D. suzukii* so dangerous compared to most Drosophila species, which feed on rotting fruits [18], is the robust, sclerosed, and serrated ovipositor, which allows them to penetrate the fruit’s epidermis [16,17,19,20,21,22]. Damage is mainly caused by larval feeding, resulting in the degradation of fruits [23], making them unmarketable [24]. Moreover, these oviposition wounds provide access to secondary fruit-feeding organisms such bacterial and fungal pathogens or other pests, resulting in additional losses [17,25,26]. Finally, fruit deterioration caused by SWD can increase its vulnerability to attacks by other drosophilid species [17]. In addition, over 50 wild host plants have been identified in the US and Europe, offering the pest a large reservoir of alternative hosts throughout the seasons [27,28].

Despite their environmental and economic drawbacks, insecticides remain the most widely used method for managing this pest [29,30,31]. Currently, SWD management relies heavily on the application of broad-spectrum insecticides, including organochlorines, organophosphates, carbamates, pyrethroids, diamides, and spinosyns [32,33]. However, most of these face increased restrictions by authorities due to their health and environmental impacts [34,35,36,37,38]. Additionally, these insecticides can have negative effects on non-target organisms, including mammals and beneficial arthropods. As their application is limited to managed crops, reinfestations often occur shortly after treatment from the surrounding vegetation [18,30]. The effectiveness of insecticides is increasingly limited due to the emergence of resistance, driven by repeated exposure, short generation time, and high fecundity [39,40,41].

Due to both the loss of efficacy and environmental unsustainability of traditional spray applications, environmentally friendly and cost-effective strategies are necessary for the control of this pest. Biorational, physical, and other non-chemical methods should be prioritized over chemical approaches if they offer effective pest control [18,42]. An integrated pest management (IPM) strategy fosters sustainable control over SWD by employing a combination of tactics, such as biological and behavioral controls, habitat manipulation, cultural practices, and the use of resistant plant varieties [18,41,43]. These combined approaches aim to reduce pest populations while minimizing reliance on chemical control.

IPM strategies not only effectively manage pests but also meet consumer demand for high-quality fruits with reduced pesticide applications [44]. Physical exclusion using fine-mesh netting, which can be installed on structures, has proven effective in reducing *D. suzukii* infestation [45,46,47,48,49,50], thereby limiting the need for pesticide use compared to open-field plots without any covering [45,46,48,49,50,51]. The increasing use of fine-mesh netting is driven by the introduction of exotic pests, such as SWD, which are difficult to control with pesticides, as well as increasing social pressure to reduce chemical inputs in agriculture and concerns about the long-term sustainability of chemical-based control methods [51,52].

Some drawbacks of using insect netting are related to the airflow resistance that affects the microclimate and the higher occurrence of disease. However, given the relatively large body size of this pest, this drawback is less important compared to other cases, such as the fight with physical barriers against the whitefly *Bemisia tabaci* or the thrips *Frankliniella occidentalis*, which need nets with much smaller holes. Crop isolation involves other issues, such as insufficient pollination and a decline in the population of natural enemies [53]. In addition, the adoption of insect netting implies both material (nets and support structures) and labor (installation and removal) cost [54]. Marketable fruit yields tend to be higher when fruits are protected under fine-mesh netting [49,51,55], and fruit quality is comparable to open-field plots [47,50,56].

Given the increasing use of insect-proof nets as a key control method [57,58,59] in combination with other measures worldwide, our aim is to evaluate the efficacy of nine commercial nets against *D. suzukii* to identify the optimal hole dimensions. We developed an experimental protocol under laboratory conditions to assess net performance while also simulating field conditions by analyzing the effects of airflow and temperature on efficacy. Additionally, since several studies have shown that temperature affects fly body size [60,61], we examined the variation in body size within a lab-reared population across a temperature range. This is important because body size can significantly impact the effectiveness of insect netting. Our hypothesis is that higher air velocity and temperatures will result in lower efficacy. Additionally, we propose that net efficacy is not a fixed value, as it is influenced by multiple factors, including pest morphometry.

## 2. Materials and Methods

### 2.1. Fine-Mesh Netting and Geometrical Characterization Fabrics

The structure of protection nets is determined by two sets of threads (weft and warp), which interweave perpendicularly and are typically made of HDPE [62]. The arrangement of threads in each direction leads to a predominantly rectangular shape for each hole, as the threads in the warp are typically closer together than those in the weft. The density of the threads in each direction is defined by the number of threads per unit length. Furthermore, the thickness of the threads is another variable that influences the geometry of the screen [63].

A set of nine commercial protective nets with the above-described characteristics was selected to evaluate their efficacy against *D. suzukii* under various conditions. The hole width *L_px_* (separation between warp threads) ranges from 638 to 1407 μm, and the hole length *L_py_* (separation between weft threads) ranges from 742 to 1535 μm, chosen based on previous studies [64].

The geometrical characteristics of the fabrics were obtained using the methodology described in a previous study [65]. This process involves the use of the Euclides v1.4 software, which analyses digital images captured via a microscope or stereomicroscope. In this case, a Motic B3 microscope equipped with a Moticam 2 digital camera (Motic, Barcelona, Spain) was used. The microscope, calibrated at 4× objective, yields a ratio of 5.54 μm per pixel (μm px^−1^). The software identifies the coordinates of the vertices that define the holes within the porous matrix under analysis. From these data points, it calculates all the key geometric parameters for the textile: hole width *L_px_* and length *L_py_*, number of threads per unit length ρ_x_ × ρ_y_ (thread density), thickness of the weft *D_hx_* and warp threads *D_hy_*, hole surface area A_2D_, and porosity φ. The thickness *t_t_* of the textiles was measured using a digital micrometer (model 3050T, Baxlo, Barcelona, Spain), following the protocol of the ISO 5084:1996 standard [66].

### 2.2. Experimental Setup and Calculation of Protective Net Efficacy

The efficacy of fine-mesh netting against SWD was assessed under laboratory conditions with uncontrolled temperature conditions, which fluctuated naturally according to outdoor environmental conditions. During the testing period, temperatures ranged between 20.4 and 28.7 °C. An experimental device [67,68] was employed to determine the exclusion capacity of the nets in both still air and at varying air velocities (Figure 1). The device consisted of a transparent PVC tube with an 11 cm diameter, divided into three chambers. The first and second chambers were connected using a flanged joint, between which the textile sample under analysis was inserted (Figure 1). The third chamber was connected to the second with another flanged joint that was fitted with a very dense fabric that allowed air to flow while preventing insects from passing from the second to the third chamber. A hot wire thermo-anemometer (model HD29V37TC12, Delta OHM, Padua, Italy) was inserted in the third chamber to measure air velocity and temperature. Additionally, a DC axial fan (4715KL-04W-B40 NBA-MAT, Minebea Co., Ltd., Tokyo, Japan) was installed at the end of this chamber to create airflow within the device. The experimental setup was controlled using the Boreas v.1.3 software, which allowed the test parameters to be set and the air velocity (kept constant throughout each test) and temperature to be recorded.

Approximately 60 flies were introduced into the first chamber of the device through a small opening in each repetition, and the chamber was immediately covered with black plastic to block out light. The second chamber was illuminated with a fluorescent tube (Philips TL-D 36W/54–765, Amsterdam, The Netherlands), which served as a visual attractant, encouraging the flies to move from the dark first chamber to the illuminated second chamber by attempting to pass through the tested net (Figure 1).

Each test lasted 24 h. After this period, the insects were anesthetized via an overdose of chloroform for easier handling, and the number of insects in the first and second chambers was counted. To ensure statistical reliability, each test was performed in triplicate. Based on these counts, the exclusion coefficient of the tested nets was calculated. The efficacy ε of a protective net was determined using the following equation:(1)$$\varepsilon \left(\%\right)=\frac{n_{1}}{n_{1}+n_{2}}\times 100$$
where *n*_1_ is the number of insects remaining in the first chamber and *n*_2_ is the number of insects that passed through the net into the second chamber.

### 2.3. Mass Rearing of D. suzukii Under Laboratory Conditions

The adults of *D. suzukii* were obtained from pupae samples supplied by the Instituto Valenciano de Investigaciones Agrarias (IVIA, Valencia, Spain) and by the Instituto de Investigación y Tecnología Agroalimentarias (IRTA, Catalonia, Spain). Petri dishes were placed in boxes with dimensions of 37 × 28 × 18 cm^3^, with a piece of non-woven fabric affixed to the cut-out section of the lid to allow ventilation inside the box and prevent the flies from escaping. The flies were reared and completed their life cycle on an artificial diet following the protocol described in [50]. The temperature in the laboratory at the University of Almeria Campus (Almeria, Spain, 36°49′36″, −2°24′36″) varied between approximately 21.0 and 28.5 °C, and the relative humidity ranged between 60 and 70% during the period from October 2022 to July 2023. The laboratory temperature was not controlled and fluctuated according to outdoor environmental conditions. The photoperiod in the laboratory was set at 16:8 h (light:dark). The flies were captured using an entomological aspirator (model E7081, Watkins & Doncaster, Leominster, UK) and then introduced into the above-described test device or anesthetized for morphometric measurements.

### 2.4. Morphometric Analysis and Insect Sample Size Estimation

The selection of fine-mesh netting as a physical protection method against a specific pest requires the geometric characterization of the textiles, a morphometric study of the pest, and a framework where these two factors interact. The morphological measurements of individual specimens were obtained from digital images captured using an Olympus SZ6 stereomicroscope (Olympus Corporation, Tokyo, Japan) equipped with a Moticam 2 digital camera (Motic, Barcelona, Spain) and the Motic Images Plus 2.0 software (Motic). Initially, the microscope was calibrated to determine the equivalent metric value for each pixel.

To facilitate handling, captured flies were placed in a Petri dish and anesthetized with a piece of cotton soaked with a few drops of chloroform. Once anesthetized, groups of five individuals were arranged on a slide for imaging. From these images, we measured the body length, the thorax width, and the abdomen width, both in dorsal view and lateral view for each analyzed insect (Figure 2).

To estimate the population means for body segment (tagma) sizes and length, we used the following expression to determine the required sample size *n* [69]:(2)$$n={\left[\frac{{\sigma \;z}_{\alpha /2}}{E}\right]}^{2}$$
where *σ* represents the population standard deviation, *E* is the allowable margin of error, and *z*_α/2_ is the critical value based on the desired confidence level (*z*_α/2_ = 1.96 for a 95% confidence level in a normal distribution). Allowing for an error margin of 10 μm and estimating the population standard deviation at 120 μm, the required sample size *n* would be greater than 553 individuals. This sample size ensures that the morphometric analysis has statistical validity.

### 2.5. Statistical Analysis

All statistical analyses were performed using a Jupyter Notebook (Project Jupyter) in Microsoft Visual Studio Code (Microsoft, Redmond, WA, USA). The data were processed and analyzed using Python (version 3.12.2) programming language with the following libraries:Pandas for data manipulation and organization;Numpy for numerical computations;Matplotlib.pyplot and seaborn for data visualization and plotting;Scipy.stats and statsmodels.formula.api (ols) for statistical analysis, including normality tests and ordinary least squares regression;Statsmodels.api for general statistical models;Statsmodels.stats.multicomp (pairwise_tukeyhsd) for post-hoc multiple comparison tests, such as Tukey’s honest significant difference (HSD) test.

The analyses conducted included one-way and two-way ANOVAs, frequency histograms, multiple linear regression, and Tukey’s HSD tests to evaluate the effects of different factors on the measured variables and identify significant differences between groups.

## 3. Results

### 3.1. Morphometric Analysis of D. suzukii

The total sample consisted of 572 SWD individuals, comprising 262 males and 310 females, used for the morphometric measurements. Differences in sample sizes between males and females are justified by the species’ sex ratio variation. The morphometric data were analyzed across a temperature range of 21.1 to 28.2 °C. To facilitate more detailed analysis, the data were divided into two temperature categories: ≤24.1 °C and >24.1 °C. This division aimed to balance the number of measurements within each range, with 24.1 °C serving as an approximate midpoint of the entire temperature interval.

Table 1 presents values for thorax, abdomen, and total body lengths across the laboratory-reared SWD population, with measurements taken from both dorsal and lateral views. The table also differentiates between the two temperature ranges (≤24.1 °C, labeled as ↓T, and >24.1 °C, labeled as ↑T) and provides mean values for each group including overall means ($\bar{T}$) for the entire temperature range (21.1–28.2 °C) to offer a comprehensive view of size variation under the full spectrum of conditions.

The data indicate that females are larger than males, with the thorax being the largest body region (tagma) for both sexes. In the dorsal and lateral views, females have a larger thorax (1013 µm and 1050 µm) compared to males (907 µm and 918 µm). These measurements suggest a more developed thoracic structure in females, but the difference between the two views is not as pronounced, indicating a relatively symmetrical thorax shape, which is important when considering the selection of mesh hole shape.

The abdomen of SWD exhibits clear sexual dimorphism, with females displaying larger dimensions than males in both dorsal and lateral views. In the dorsal view, the female abdomen measures 1001 µm compared to 894 µm for males, while in the lateral view, females measure 766 µm versus 660 µm for males. This suggests that females possess a broader and more developed abdominal structure, potentially linked to their reproductive role. In both sexes, the abdomen is wider than it is deep, indicating a more flattened geometric shape. However, its smaller size compared to the thorax makes this geometric characteristic less significant.

The size differences between males and females are more pronounced in body length than in thorax and abdomen dimensions.

The data also indicate a clear influence of temperature on the body dimensions of SWD, with individuals reared at lower temperatures (21.1–24.1 °C) exhibiting larger body sizes compared to those reared at higher temperatures (24.1–28.2 °C). This trend is consistent across both sexes and all measured traits (body length and thorax and abdomen dimensions). For example, the total body length of males and females is significantly larger in the lower temperature range, with males averaging 2350 µm and females 2804 µm at 21.1–24.1 °C, compared to 2180 µm and 2615 µm at 24.1–28.2 °C. Similarly, the thorax (both dorsal and lateral views) and the abdomen (dorsal view) decrease in size as temperature increases, with larger values recorded at the lower temperature range. However, abdomen height in the lateral view *A_L_* is unique in that only females show an increase in size with rising temperatures, while males experience a decrease.

This reduction in size at higher temperatures is more pronounced in males. For instance, male thorax width in the dorsal view decreases from 964 µm to 842 µm between the lower and higher temperature ranges (a 12.7% reduction), while in females, it decreases from 1074 µm to 959 µm (a 10.7% reduction). A similar pattern is observed for thorax lateral measurements and abdomen dimensions.

The two-way ANOVA results presented in Table 2 offer insights into how temperature (T) and sex influence the body dimensions of SWD, as well as whether there is an interaction between these two factors.

According to the results shown in Table 2, both temperature and sex have highly significant effects on body length *L*, with very low *p*-values (<0.001) indicating strong evidence that both factors independently affect this trait. Specifically, sex has a much larger effect on body length (F = 384.05), confirming that females are significantly larger than males. However, the interaction between temperature and sex is not significant (*p* = 0.699), meaning that the effect of temperature on body length is similar for both sexes.

Temperature and sex also significantly affect thorax width in the dorsal view *T_D_*, with both factors showing very low *p*-values (<0.001). Sex significantly influences thorax width *T_D_*, with an F-value of 237.80, while temperature also has a notable effect on this trait, with an F-value of 260.21. Similarly to body length, the interaction between temperature and sex is not significant (*p* = 0.624), indicating that the influence of temperature on this trait is consistent across males and females.

For the thorax in the lateral view *T_L_*, both temperature and sex are significant (*p* < 0.001), and there is now a significant interaction between temperature and sex (*p* = 0.003). This suggests that the effect of temperature on thorax size differs between males and females, with one sex being more sensitive to temperature changes than the other. Specifically, temperature has a greater influence on males, who exhibit a pronounced decrease in thorax size under higher temperatures compared to their lower temperature measurements.

Regarding abdomen width in the dorsal view *A_D_*, temperature and sex significantly affect this tagma (*p* < 0.001), with no significant interaction (*p* = 0.861), suggesting that the size differences between males and females are stable across different temperatures.

In contrast to the other traits, temperature does not have a significant effect on abdomen height in the lateral view *A_L_* (*p* = 0.706), while sex has a strong influence (F = 82.63, *p* < 0.001). There is a significant interaction between temperature and sex (*p* < 0.001), indicating that temperature affects males and females differently in this trait. Specifically, male abdomen height decreases slightly from 680 μm at lower temperatures to 638 μm at higher temperatures, while females exhibit an increase from 743 μm to 786 μm. This interaction suggests that the abdomen’s response to temperature changes is more complex in the lateral view, possibly due to sex-specific differences in abdominal development or function.

The two-way ANOVA revealed significant effects of temperature, sex, and their interaction on several morphological traits of *D. suzukii* (Table 2). To further explore these differences, a Tukey’s HSD post-hoc test was conducted to compare the mean values of each trait between temperature levels and sexes. The results of these pairwise comparisons are summarized in Table 3.

The Tukey HSD test revealed (Table 3) that both temperature and sex had a significant effect on all morphological traits of *D. suzukii*, except for A_L_ when comparing the low and high temperature intervals. In this specific case, the difference in means was −11.3 µm, with an adjusted *p*-value of 0.365 (Table 3), indicating that the temperature range considered did not lead to significant changes in the lateral abdominal dimension.

Given the significant interaction between temperature and sex detected using the two-way ANOVA (Table 2), a Tukey’s HSD post-hoc test was performed to assess pairwise differences between temperature–sex combinations for each morphological trait. The results of these comparisons are presented in Table 4.

The Tukey HSD test assessing the interaction between temperature and sex showed significant differences in all morphological traits of SWD, except in four specific comparisons (Table 4). For each group (trait), the greatest differences are indicated by the absolute values in the third column (mean difference) of Table 4. No statistically significant differences were detected for *T_D_*, *T_L_*, *A_D_*, and *A_L_* when comparing females at high temperatures with males at low temperatures. This pattern suggests that, despite the overall influence of temperature and sex, the combination of these specific groups results in similar trait measurements, because females reared at high temperatures tend to be smaller and males reared at low temperatures tend to be larger. Figure 3 shows a series of frequency histograms with corresponding normal distributions, providing a comparative analysis of size distributions for key morphological traits of *D. suzukii*. The first row of histograms presents the overall size distributions of thorax (dorsal view, *T_D_*; lateral view, *T_L_*) and abdomen (dorsal view, *A_D_*; lateral view, *A_L_*) measurements across the entire temperature range (21.1–28.2 °C). The second row compares female size distributions across two temperature intervals: 21.1–24.1 °C (↓T) and 24.1–28.2 °C (↑T). Similarly, the third row illustrates the same comparison for male specimens.

In the histograms (Figure 3) representing the body dimensions of SWD, clear effects of both temperature and sex are observed. Across all traits, females consistently exhibit larger sizes than males. The size difference between sexes is most pronounced in thorax dimensions, with females showing a tighter clustering around the mean, while males display more variability, particularly in thorax and abdomen lateral views.

### 3.2. Geometrical Characteristics of Protective Nets

Table 5 summarizes the geometric characteristics of the nine exclusion nets included in this study. The parameters considered include the thread density in both weft (ρ_x_) and warp (ρ_y_), hole dimensions (*L_px_* and *L_py_*), and the shape factor **ψ** (*L_px_*/*L_py_*), as well as the thickness of the threads in weft (*D_hx_*) and warp (*D_hy_*), mean hole surface area (*A_2D_*), mesh thickness (*t_t_*), and porosity (φ). The nets differ not only in their geometric properties but also in thread color (crystal, black, and green). Although thread color is not a variable studied in this research, it is included in the table due to its potential influence on some results.

The geometric characteristics of the nine insect-proof nets selected for testing against SWD (Table 5) reveal important distinctions that may influence their effectiveness. The nets vary in thread densities (ρ_x_ × ρ_y_), with values ranging from 5.6 × 5.7 to 9.6 × 10.0 threads cm^−2^. This variation, along with the thread thicknesses, directly affects the hole sizes (*L_px_* and *L_py_*), which range from 638 to 1407 μm in width and 742 to 1535 μm in length. The shape factor **ψ** (*L_px_*/*L_py_*) also differs, with some nets having more elongated holes (ratios as low as 0.600), while others have almost square holes (ratios close to 1.0).

The thread thickness (*D_hx_* and *D_hy_*) shows relatively small variation, between 280 and 360 μm, with no clear correlation between thread thickness and hole size. This reflects a rigid design strategy by manufacturers, who do not consider the relationship between thread density and thread thickness to achieve the desired hole sizes while optimizing fabric porosity. Just as with hole sizes, thread thickness impacts the 2D surface area of the holes *A_2D_*, which varies significantly from 0.530 to 2.027 mm^2^. The thickness of the nets *t_t_* ranges from 370 to 568 μm, reflecting how tightly packed the weft and warp threads are. This thread tightness plays a crucial role in determining the real (3D) hole opening. However, this is an important parameter that manufacturers do not take into account at all. Additionally, the porosity (φ) ranges from 49.8% to 68.2%, indicating varying levels of air and light transmission. These factors are critical for both crop protection and maintaining adequate environmental conditions for plant growth.

### 3.3. Efficacy of Protective Nets in Excluding D. suzukii

The results of efficacy for each protective net are shown in Table 6. For each fabric, the results are provided for the temperature range between 20.4 and 25 °C (↓T), between 25 and 28.7 °C (↑T), and the entire temperature range ($\bar{T}$). Each cell in this table consists of two values [ε (%); T (°C)]: efficacy and the temperature at which the trials were conducted. Each efficacy value is the mean of three repetitions performed for each air velocity level (0, 1.5, and 3 m s^−1^), and therefore, the temperature is also the mean of three sets of records. Consequently, the rows labeled $\bar{T}$ represent the average of six repetitions. Using the same logic, the sex ratios of the original sample introduced into the test device and the sample present in the second chamber of the device (the SWD sample that was able to pass through the mesh) are also presented.

The efficacy of nets 1 and 2 (Figure 4) was very low under calm conditions, which is why tests at higher speeds were not conducted. Nets 6, 7, and 9 were fully effective against SWD at both high air velocities and high temperatures, while nets 3, 5, and 8 showed very high efficacy values. According to the results, hole widths around 700 μm are required (ψ = 0.792) to achieve very high protection against SWD, as is the case with mesh 8, which provides very high but not complete protection. Based on the results, a hole width of approximately 715 μm can be acceptable (nets 8 and 9) as long as the hole length is related to a shape factor not less than 0.8. Other interesting results are obtained with hole widths below 700 μm (nets 6 and 7), which allow for shape factors as low as 0.6–0.7, while improving porosity values above 53%.

Nets 6 and 7 (completely effective) have hole widths below 700 μm and hole lengths around 1000 μm, with form factors ψ of 0.699 and 0.600, respectively. Net 9 (also completely effective) has a hole width of 715 μm and a pore length of 742 μm, with a form factor of 0.964. The design of nets 6 and 7 appears to be more suitable, as they provide complete protection against SWD and their porosity is higher than that of net 9, which will undoubtedly improve the crop conditions.

Net 3 has larger hole dimensions (816 × 1333) µm^2^ compared to net 4 (778 × 1280) µm^2^. The form factors ψ of both nets are quite similar, 0.608 and 0.612, respectively. However, net 3 (with larger hole size) achieved better efficacy against SWD, which contradicts the logical expectation based on hole size alone. A difference between the two nets lies in the color of the threads: net 3 is made with crystal-colored threads, whereas net 4 has green threads. Therefore, thread color could be a relevant factor to consider in relation to the visual attractiveness of the net to this species.

The results (Table 6) of the efficacy tests reveal notable patterns influenced by both temperature and airflow velocity. For tests performed within the lower temperature range (21.1–24.1 °C, ↓T), the efficacy values were significantly higher compared to those in the higher temperature range (24.1–28.2 °C, ↑T). For example, net 3 exhibited an efficacy of 100% at 23.8 °C under 0 m/s airflow and 91.0% at 25.4 °C, demonstrating the nets’ effectiveness in blocking the passage of insects at lower temperatures. In contrast, the efficacy decreased at higher temperatures, suggesting that *D. suzukii* may be more active (or smaller) and capable of passing through the nets as temperatures rise. We use the term “smaller” because higher temperatures are associated with smaller flies.

Regarding airflow velocity, the efficacy measurements varied with different airflow velocities (0, 1.5, and 3.0 m/s). For example, in net 3 at ↓T, the efficacy was 100% at 0 m/s but dropped to 86.0% at 3.0 m/s. In the case of net 4, the efficacy decreased significantly when the air velocity increased from 1.5 to 3 m s^−1^, and this is one of the fabrics with the greatest thickness *t_t_* (566 μm). This trend is consistent across most nets, indicating that increased airflow velocity may facilitate the passage of *D. suzukii*, potentially due to increased pressure or turbulence disrupting the nets’ barrier protection.

The sex ratios of the initial samples compared to those that crossed the nets indicate a trend towards a reduction in the number of females that successfully traversed the exclusion barriers. For example, the initial ratios show that females were consistently more prevalent (e.g., ratios like 1:1.12 and 1:1.24), while the ratios of those that crossed the nets indicate a decline in the proportion of females, such as 1:0.89 and 1:0.82 for nets 7 and 9, respectively. The selective permeability of the nets favors the passage of smaller males, leading to a shift in the sex ratio among those that successfully cross.

A multiple regression analysis was conducted to assess the influence of the variables involved in the experimental design on the efficacy of the insect-proof meshes. External variables, such as air velocity and temperature, and mesh-related variables, including hole width *L_px_*, hole length *L_py_*, mesh thickness *t_t_*, and shape factor **ψ**, have been considered. The results from this initial attempt to model the influence of the variables on the efficacy of the protective meshes showed an R^2^ = 0.855 and indicated that neither hole length *L_py_* and mesh thickness *t_t_* nor the shape factor *L_px_*/*L_py_* had a significant influence on mesh efficacy, with *p*-values (0.627, 0.932, and 0.393, respectively) exceeding 0.05. However, we know from other studies [70] and from the results of this work (see the results for nets 8 and 9) that hole length *L_py_* is an influential variable in the efficacy of exclusion nets. Nonetheless, the experimental design of this study did not aim to investigate this factor specifically, in part due to the limited commercial availability of meshes, which made it impossible to control other geometric variables while varying *L_py_*. Therefore, the analysis was repeated excluding these variables, and the results are presented in Table 7.

This multiple linear regression explains 84.7% of the variability in mesh efficacy due to air velocity *u*, air temperature *T*, and hole width *L_px_*. Considering that mesh efficacy can only vary between 0 and 100 (when expressed as a percentage), we express the result of the linear regression model as shown below.(3)$$\varepsilon \left(u,T,L_{px}\right)=\left\{\begin{array}{lll} 0, & if & -1.64u-0.97T-0.15L_{px}+226.98<0\\ -1.64u-0.97T-0.15L_{px}+226.98, & if & 0\leq -1.64u-0.97T-0.15L_{px}+226.98\leq 100\\ 100, & if & -1.64u-0.97T-0.15L_{px}+226.98>100 \end{array}\right.$$

We do not find it coherent to conduct an ANOVA with the temperature intervals to support the results of the multiple regression analysis, as the experimental reality is that the temperature varied continuously between 20.4 and 28.7 °C. Therefore, establishing a low and high temperature range serves the purpose of presenting results and understanding the phenomenon but does not align with the true nature of the ANOVA analysis. However, the air velocities *u* are categorized (0, 1.5, and 3 m s^−1^), and thus, in this case, the ANOVA analysis is appropriate. The data from this analysis support the results obtained in the multiple regression analysis and highlight the influence of air velocity on efficacy values (F-value = 6.818 and *p*-value = 0.002).

To analyze whether the means of the air velocity groups show statistically significant differences, we performed a Tukey’s test, the results of which are presented in Table 8. Significant differences are observed between the 0 and 1.5 m s^−1^ groups (and, of course, between the 0 and 3 m s^−1^ groups as well). However, the analysis does not allow us to reject the null hypothesis of equal means between the 1.5 and 3.0 m s^−1^ groups, which may have important practical implications.

The greatest impact on efficacy appears to occur when moving from zero velocity (0 m s^−1^) to a moderate velocity (1.5 m s^−1^). Beyond this point, further increasing the velocity (up to 3 m s^−1^) does not lead to significant differences in efficacy, although the general trend observed in the regression model remains negative (efficacy decreases as air velocity increases). As previously mentioned, mesh 4 is an exception to this trend, revealed by the Tukey analysis, as in this case, there is a drastic drop in efficacy when the velocity increases from 1.5 to 3.0 m s^−1^. Perhaps the combination of hole width (*L_px_* = 778 μm), length (*L_py_* = 1280 μm), the shape factor (*L_px_*/*L_py_* = 0.608), and net thickness (*t_t_* = 566 μm) creates a confluence of factors that leads to this drastic drop in efficacy.

## 4. Discussion

### 4.1. Morphometric Analysis of D. suzukii

Several studies have highlighted the differences in body size between male and female *D. suzukii*, often attributing these variations to environmental factors and sexual dimorphism [41,60,61,71,72]. SWD body size is strongly affected by temperature, with 13 °C and 28 °C causing the largest and the smallest body sizes, respectively [73]. Research shows that females are typically larger than males, especially in terms of body and wing sizes. In this work, we did not measure wing size, which has been considered in other research examining the phenotypic plasticity of *D. suzukii*. However, the orientation and flexibility of the wings relative to the body do not seem to significantly hinder the flies from passing through the mesh holes, although this is a point that should be further investigated.

The results obtained align with the general biological principle that insects reared at lower temperatures tend to grow larger, as slower development rates at cooler temperatures often allow for more prolonged growth periods. In contrast, at higher temperatures, faster developmental rates may lead to smaller body sizes. The greater size reduction observed in males suggests that they may be more sensitive to temperature changes than females, although both sexes exhibit this pattern, particularly within the temperature interval we tested.

According to the size variability shown in the frequency histograms (Figure 3), the larger body sizes observed in individuals developing at lower temperatures are evident in both sexes, though the effect is more pronounced in females. The distribution of body sizes in the higher temperature group also shows greater variability, suggesting that elevated temperatures may induce more phenotypic variation. These findings align with the above-mentioned and well-established relationship between lower temperatures and larger body sizes in insects, likely due to prolonged developmental periods in cooler environments [74,75,76].

Despite the more limited temperature range in this study (we cannot refer to these as cold temperatures), we still observe similar trends in sexual dimorphism (Figure 5). This suggests that even within a narrower thermal spectrum, the phenotypic plasticity of *D. suzukii* remains evident. These findings align with broader patterns seen in the species, emphasizing the robustness of this dimorphism even under subtler environmental changes.

Regarding the topic at hand, it is crucial to select the mesh hole size based on the climatic variables, particularly temperature, of the cultivation site. Since temperature significantly influences the body size of SWD, choosing a hole size that aligns with these variations can enhance the effectiveness of the chosen solution. A protective net that is effective under cooler conditions may not perform as well under warmer temperatures, where flies tend to be smaller, highlighting the importance of tailoring netting specifications to the local climate.

### 4.2. Efficacy of Protective Nets in Excluding D. suzukii

There are very few previous laboratory studies specifically evaluating the efficacy of insect-proof netting against *D. suzukii* [52], although numerous field trials have been conducted. Comparing the results obtained in laboratory conditions with those from field trials is not straightforward, as several variables affect the efficacy in the field, including pest pressure, host crop attractiveness, and crop phenology, which are not accounted for in laboratory settings [53].

Kuesel et al. [51] reported in field trials that a 0.85 mm by 1.4 mm mesh size reduced the capture of SWD in baited traps and decreased the overall number of SWD adults emerging from harvested blackberry fruits. Additionally, field trials have shown that a 0.85 mm by 1.00 mm mesh size yielded positive results [56]. Other studies [47,55] report that using fine-mesh netting (0.6 mm × 1.0 mm) installed on structures (e.g., high tunnels) effectively reduced SWD in soft fruits compared to uncovered open-field plots. More conservative suggestions also state that hole sizes should be as small as 0.5 mm by 0.8 mm [46]. We could continue expanding the list of mesh sizes recommended by various authors based on field results [45,47,48,49,50,77], which demonstrates a certain lack of consistency in criteria (due to changing conditions that result in varying needs).

Although recent studies continue to emphasize the use of insect-proof nets as a sustainable cultural control measure, reporting promising results, unfortunately, some of these studies pay little attention to the actual geometric properties of the fabrics, providing incomplete information or relying on commercial brands that do not disclose the hole sizes of the materials used [45,48,49,50].

Considering the textile barriers used and the test conditions (air velocity and temperature), it can be stated (Table 5 and Table 6) that the maximum hole width providing full protection against SWD is 715 μm, as observed with mesh 9. This leads to an interesting observation, as mesh 8 has a nearly identical hole width (716 μm), yet its efficacy is not complete for air velocities from 1.5 m s^−1^ and for higher temperature ranges. This finding is consistent with the ’prison effect’ described by [70], suggesting that the increased hole length *L_py_* (904 vs. 742 μm), along with a lower shape factor (0.792 vs. 0.964), accounts for this loss of efficacy despite the hole width remaining the same. This observation is further reinforced by examining mesh 5, which, despite having a slightly larger hole width *L_px_* (748 μm), also has a significantly greater hole length *L_py_* (1221 μm) and does not provide full protection even under calm conditions and within the higher temperature range. However, Chouinard et al. [52] found that rectangular mesh with a low aspect ratio (0.5) had the highest exclusion efficacy in laboratory tests. While a 1.3 mm square mesh excluded 52% of females and 25% of males, a 0.9 mm by 1.8 mm rectangular mesh achieved 100% efficacy for both sexes. Although both meshes have similar surface areas, the dimensions of the hole (width and length) are very different, which likely explains the differences in efficacy more than the shape of the hole itself. We believe that comparing the efficacy of fine-mesh exclusion netting based on the shape of the hole should be performed by keeping one dimension constant and varying the others. They [52] also noted that meshes with 1.0 × 1.0 mm and 0.85 × 1.4 mm apertures nearly excluded all females, although some males passed through the latter. Their conclusions were based on thorax measurements from only ten individuals and did not specify the test temperature.

Considering the variability in body size, we recommend using body size percentiles or accounting for smaller individuals when selecting the appropriate barrier. Based on our data, we observed that females can be as small as 654 µm (*T_D_*) in the higher temperature interval, or 824 µm in the lower temperature interval. This suggests that a mesh with hole dimensions slightly smaller than these values may be necessary.

The combination of temperature and airflow velocity appears to significantly influence the efficacy of the insect nets against *D. suzukii*. The data suggest that lower temperatures and higher air velocities are associated with increased net effectiveness. Regarding temperature, Hamby et al. [78] state that during the summer months, SWD adults are most active at temperatures ranging between 15 and 20 °C, with activity decreasing at temperatures above and below this range. Our results align with this observation regarding increased insect activity (although the loss of net efficacy may be attributed to smaller body sizes). However, they differ from the specific temperature range provided by the authors. In our study, within the higher temperature range (20.4–28.7 °C), net effectiveness was generally lower, which we attribute to increased insect activity, smaller body sizes, or both. Additionally, the observed sex ratio changes may have implications for understanding the dynamics of pest control using physical barriers.

## 5. Conclusions

This study provides insights into both the morphological variation in *D. suzukii* and the efficacy of protective fine meshes as a sustainable management solution against this pest. Our morphometric analysis revealed significant sexual dimorphism. Additionally, temperature had a marked effect on body size. These findings highlight the importance of considering pest morphology when designing and selecting insect-proof screens, as body size can influence the effectiveness of physical barriers.

Regarding the efficacy of the nine tested nets, our results demonstrated that net performance is highly sensitive to changes in environmental conditions, particularly air velocity and temperature. At higher temperatures and air velocities, nets with smaller hole sizes will be needed. This variability challenges the common assumption that the efficacy of insect netting is constant. By identifying the interplay between airflow, temperature, and net design, our study highlights the importance of optimizing hole dimensions based on these factors to enhance protection against *D. suzukii*.

Overall, these findings support the use of insect netting as a sustainable alternative to chemical control. Our research indicates that it is possible to completely prevent SWD if an appropriate mesh hole size is chosen.

Future research should broaden the temperature range studied and investigate the impact of hole shape on insect exclusion. Further experiments could refine the multiple linear regression model, while field studies would help validate laboratory results. It would also be beneficial to validate laboratory results through field studies, applying the findings to real-world environmental conditions. Exploring more sustainable materials than HDPE for mesh production should be prioritized to enhance environmental sustainability [59]. Additionally, studying mesh hole size variability and the potential effect of nets on body size selection in SWD populations could provide valuable insights.

## Figures and Tables

**Figure 1 insects-16-00253-f001:**
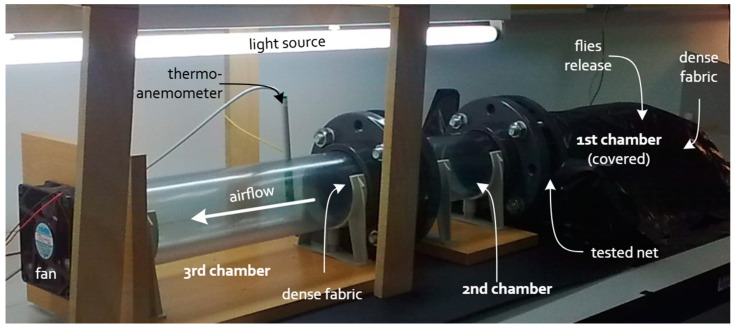
Experimental device used to assess the efficacy of protective nets, featuring a transparent PVC tube divided into three chambers and equipped with an axial fan and a thermo-anemometer for airflow and temperature control.

**Figure 2 insects-16-00253-f002:**
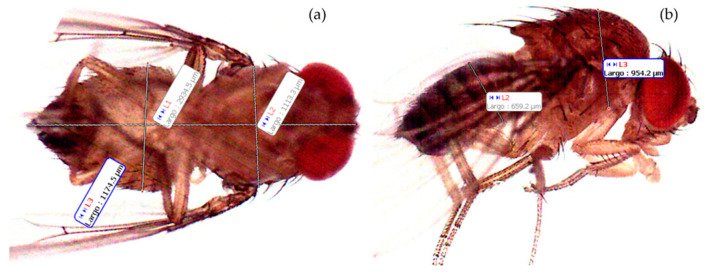
Male SWD under a microscope showing body size measurements: (**a**) ventral view and (**b**) lateral view.

**Figure 3 insects-16-00253-f003:**
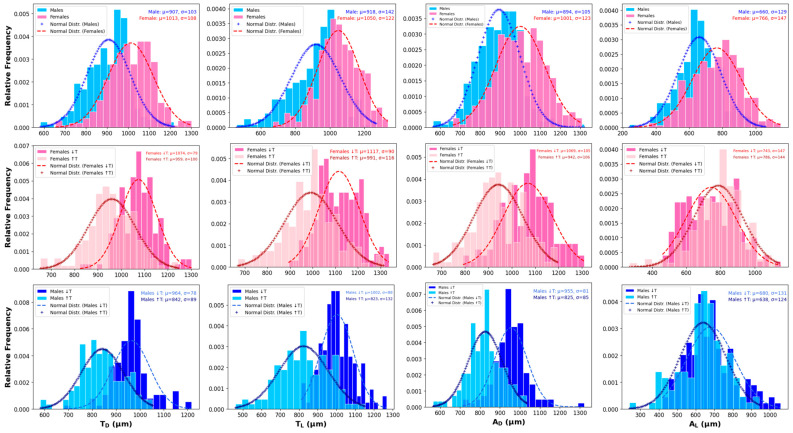
Frequency histograms of key morphological traits of SWD (thorax dorsal, *T_D_*; thorax lateral, *T_L_*; abdomen dorsal, *A_D_*; abdomen lateral, *A_L_*). The first row represents the size distributions for all specimens (

 for males and 

 for females) across the full temperature range (21.1–28.2 °C). The second and third rows compare the size distributions of females (

 at ↑T and 

 at ↓T) and males (

 at ↑T and 
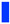
 at ↓T), respectively, within two temperature intervals (21.1–24.1 °C (↓T) and 24.1–28.2 °C (↑T)). Normal distributions are superimposed onto each histogram to illustrate the variation in body size by sex and temperature range.

**Figure 4 insects-16-00253-f004:**
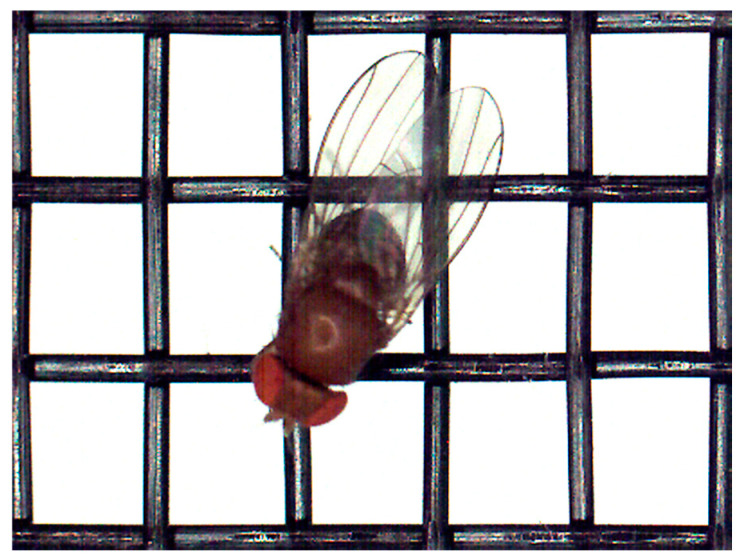
Female *D. suzukii* on protective net number 2.

**Figure 5 insects-16-00253-f005:**
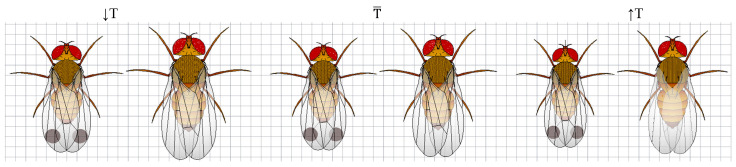
Representative male and female *D. suzukii* pairs showing body sizes across three temperature intervals: the low-temperature interval (21.1–24.1 °C, ↓T), the full temperature range (21.1–28.2 °C, $\bar{T}$), and the high-temperature interval (24.1–28.2 °C, ↑T). The figure illustrates body size variation across these intervals, showing the general trend of larger flies in the low-temperature interval and smaller flies in the high-temperature interval.

**Table 1 insects-16-00253-t001:** Morphometric characterization of SDW measured across different temperature intervals. Minimum, maximum, and mean values are provided for thorax measurements in dorsal *T_D_* and lateral *T_L_* views, abdomen measurements in dorsal *A_D_* and lateral *A_L_* views, and total body length *L* for both males and females. Data are presented for two temperature ranges: (21.1–24.1) °C (↓T) and (24.1–28.2) °C (↑T), along with overall means for the entire temperature range (21.1–28.2 °C, $\bar{T}$); sd denotes the standard deviation.

Trait	Temp. Interval	Minimum (µm)		Mean ± sd (µm)		Maximum (µm)	
		♂	♀	♂	♀	♂	♀
	↓T	1961	2293	2350 ± 224	2804 ± 247	2876	3461
*L*	$\bar{T}$	---	---	2270 ± 261	2703 ± 302	---	---
	↑T	1594	1774	2180 ± 272	2615 ± 318	2751	3212
	↓T	681	824	964 ± 78	1074 ± 79	1217	1302
*T_D_*	$\bar{T}$	---	---	907 ± 103	1013 ± 108	---	---
	↑T	583	654	842 ± 89	959 ± 100	1053	1245
	↓T	812	893	1002 ± 88	1117 ± 90	1267	1337
*T_L_*	$\bar{T}$	---	---	918 ± 142	1050 ± 122	---	---
	↑T	460	669	823 ± 132	991 ± 116	1150	1312
	↓T	713	791	955 ± 82	1069 ± 105	1321	1309
*A_D_*	$\bar{T}$	---	---	894 ± 106	1001 ± 123	---	---
	↑T	565	663	825 ± 86	942 ± 107	1034	1193
	↓T	419	460	680 ± 131	743 ± 148	1057	1150
*A_L_*	$\bar{T}$	---	---	660 ± 130	766 ± 147	---	---
	↑T	241	268	638 ± 125	786 ± 144	942	1140

**Table 2 insects-16-00253-t002:** Two-way ANOVA results for the effects of temperature (T), sex, and their interaction on the morphometric traits of *D. suzukii*.

Trait	Source	Sum of Squares	df	F	*p*-Value
	T	4.63 × 10^6^	1	63.67	<0.001
*L*	Sex	2.79 × 10^7^	1	384.05	<0.001
	T × Sex	1.08 × 10^4^	1	0.15	0.699
	Residual	4.13 × 10^7^	568	---	---
	T	2.00 × 10^6^	1	260.21	<0.001
*T_D_*	Sex	1.83 × 10^6^	1	237.80	<0.001
	T × Sex	1.85 × 10^3^	1	0.24	0.624
	Residual	4.37 × 10^6^	568	---	---
	T	3.20 × 10^6^	1	273.82	<0.001
*T_L_*	Sex	2.81 × 10^6^	1	240.88	<0.001
	T × Sex	1.02 × 10^5^	1	8.77	0.003
	Residual	6.63 × 10^6^	568	---	---
	T	2.33 × 10^6^	1	251.31	<0.001
*A_D_*	Sex	1.88 × 10^6^	1	202.90	<0.001
	T × Sex	2.83 × 10^2^	1	0.03	0.861
	Residual	5.27 × 10^6^	568	---	---
	T	2.72 × 10^3^	1	0.14	0.706
*A_L_*	Sex	1.57 × 10^6^	1	82.63	<0.001
	T × Sex	2.56 × 10^5^	1	13.42	<0.001
	Residual	1.08 × 10^7^	568	---	---

**Table 3 insects-16-00253-t003:** Tukey’s HSD test results for the effect of temperature and sex on the morphological traits of SWD. Data are presented for two temperature ranges: 21.1–24.1 °C (↓T) and 24.1–28.2 °C (↑T).

Trait	Comparison	Mean Difference (µm)	95% CI (µm)	Adjusted *p*-Value
*L*	↑T vs. ↓T	151.2	[94.0, 208.5]	0.000
	♀ vs. ♂	−432.5	[−479.3, −385.7]	0.000
*T_D_*	↑T vs. ↓T	111.2	[94.0, 128.3]	0.000
	♀ vs. ♂	−106.0	[−123.4, −88.5]	0.000
*T_L_*	↑T vs. ↓T	140.6	[119.4, 161.9]	0.000
	♀ vs. ♂	−131.1	[−152.9, −109.4]	0.000
*A_D_*	↑T vs. ↓T	120.4	[102.0, 138.8]	0.000
	♀ vs. ♂	−106.9	[−126.0, −87.9]	0.000
*A_L_*	↑T vs. ↓T	−11.3	[−35.8, 13.2]	0.365
	♀ vs. ♂	−105.8	[−128.8, −82.8]	0.000

**Table 4 insects-16-00253-t004:** Tukey’s HSD test results for the interaction between temperature and sex on morphological traits of SWD. Data are presented for two temperature ranges: 21.1–24.1 °C (↓T) and 24.1–28.2 °C (↑T).

Trait	Comparison	Mean Difference (µm)	95% CI (µm)	Adjusted *p*-Value
*L*	♀ (↑T) vs. ♂ (↑T)	−435.7	[−518.3, −353.2]	0.000
	♀ (↑T) vs. ♀ (↓T)	188.4	[109.2, 267.5]	0.000
	♀ (↑T) vs. ♂ (↓T)	−264.8	[−344.7, −184.9]	0.000
	♂ (↑T) vs. ♀ (↓T)	624.0	[538.7, 709.4]	0.000
	♂ (↑T) vs. ♂ (↓T)	170.9	[84.8, 256.9]	0.000
	♀ (↓T) vs. ♂ (↓T)	−453.2	[−535.8, −370. 6]	0.000
*T_D_*	♀ (↑T) vs. ♂ (↑T)	−117.4	[−144.3, −90.5]	0.000
	♀ (↑T) vs. ♀ (↓T)	115.3	[89.6, 141.1]	0.000
	♀ (↑T) vs. ♂ (↓T)	5.1	[−20.9, 31.1]	0.957
	♂ (↑T) vs. ♀ (↓T)	232.7	[205.0, 260. 5]	0.000
	♂ (↑T) vs. ♂ (↓T)	122.6	[94.6, 150.5]	0.000
	♀ (↓T) vs. ♂ (↓T)	−110.2	[−137.1, −83.3]	0.000
*T_L_*	♀ (↑T) vs. ♂ (↑T)	−168.0	[−201.1, −134.8]	0.000
	♀ (↑T) vs. ♀ (↓T)	125.2	[93. 5, 157.0]	0.000
	♀ (↑T) vs. ♂ (↓T)	11.0	[−21.0, 43.0]	0.811
	♂ (↑T) vs. ♀ (↓T)	293.2	[259.0, 327.3]	0.000
	♂ (↑T) vs. ♂ (↓T)	179.0	[144.5, 213.5]	0.000
	♀ (↓T) vs. ♂ (↓T)	−114. 2	[−147.3, −81.0]	0.000
*A_D_*	♀ (↑T) vs. ♂ (↑T)	−116.8	[−146.3, −87. 3]	0.000
	♀ (↑T) vs. ♀ (↓T)	126.7	[98.4, 154.9]	0.000
	♀ (↑T) vs. ♂ (↓T)	12.7	[−15.8, 41.2]	0.661
	♂ (↑T) vs. ♀ (↓T)	243.5	[213.0, 273.9]	0.000
	♂ (↑T) vs. ♂ (↓T)	129.500	[98.8, 160.2]	0.000
	♀ (↓T) vs. ♂ (↓T)	−114.0	[−143.5, −84.5]	0.000
*A_L_*	♀ (↑T) vs. ♂ (↑T)	−148.1	[−190.4, −105.8]	0.000
	♀ (↑T) vs. ♀ (↓T)	−43.3	[−83.9, −2.8]	0.031
	♀ (↑T) vs. ♂ (↓T)	−106. 4	[−147.3, −65.5]	0.000
	♂ (↑T) vs. ♀ (↓T)	104.7	[61.0, 148.4]	0.000
	♂ (↑T) vs. ♂ (↓T)	41.7	[−2.3, 85.7]	0.071
	♀ (↓T) vs. ♂ (↓T)	−63.0	[−105.3, −20.7]	0.001

**Table 5 insects-16-00253-t005:** Geometric characteristics of the protective nets: thread color, average thread density in the weft ρ_x_ and warp ρ_y_ directions, average hole width *L_px_*, average hole length *L_py_*, the shape factor *L_px_*/*L_py_*, average thread thickness in the weft *D_hx_* and warp *D_hy_* threads, surface area *A_2D_*, textile thickness *t_t_*, and porosity φ.

Net	Thread Color	ρ_x_ × ρ_y_ (Threads cm^−2^)		*L_px_* ± sd (μm)	*L_py_* ± sd (μm)	*L_px_/L_py_* (ψ) (μm μm^−1^)	*D_hx_* ± sd (μm)	*D_hy_* ± sd (μm)	A_2D_ (mm^2^)	t_t_ ± sd (μm)	φ (%)
		**Commercial**	**Measured**								
1	crystal	6 × 6	5.6 × 5.7	1407 ± 35	1441 ± 47	0.976	360 ± 12	353 ± 12	2.027 ± 0.084	498 ± 4	64.0
2	black	6 × 7	5.5 × 6.9	1175 ± 26	1535 ± 49	0.765	280 ± 9	281 ± 13	1.804 ± 0.069	510 ± 7	68.2
3	crystal	6 × 9	6.1 × 8.8	816 ± 24	1333 ± 24	0.612	314 ± 16	315 ± 16	1.087 ± 0.037	530 ± 5	58.4
4	green	6 × 9	6.2 × 9.2	778 ± 82	1280 ± 70	0.608	324 ± 53	313 ± 47	0.992 ± 0.113	566 ± 11	56.9
5	crystal	6 × 9	6.6 × 9.4	748 ± 42	1221 ± 25	0.613	302 ± 16	319 ± 14	0.913 ± 0.056	568 ± 8	56.2
6	crystal	8 × 10	7.7 × 10.1	696 ± 22	996 ± 15	0.699	305 ± 8	298 ± 10	0.693 ± 0.027	370 ± 5	53.6
7	crystal	7 × 10	7.4 × 10.6	638 ± 26	1063 ± 38	0.600	304 ± 7	289 ± 10	0.678 ± 0.037	508 ± 4	53.3
8	crystal	8 × 10	8.1 × 9.5	716 ± 45	904 ± 21	0.792	328 ± 12	341 ± 17	0.648 ± 0.045	468 ± 8	49.8
9	crystal	10 × 10	9.6 × 10.0	715 ± 25	742 ± 30	0.964	305 ± 17	285 ± 9	0.530 ± 0.028	458 ± 4	50.6

**Table 6 insects-16-00253-t006:** Mean efficacy ε of the fine-mesh netting tested, along with the mean temperature in the lower temperature range (↓T) and higher temperature range (↑T), as well as the overall mean temperature of the above ($\bar{T}$) for each air velocity level (0, 1.5, and 3.0 m s^−1^). Efficacy results are also presented with the mean sex ratios obtained in each set of tests.

Net	T	ε (%); *T* (°C)			Sex Ratio (♂:♀)					
					Initial Sample			Crossed the Net		
		*u* (m s^−1^)			*u* (m s^−1^)					
		0	1.5	3.0	0	1.5	3	0	1.5	3
	↓T	13.5; 21.4	---	---	1:1.12	---	---	1:1.05	---	---
1	$\bar{T}$	10.5; 24.2	---	---	1:0.93	---	---	1:0.86	---	---
	↑T	7.6; 27.0	---	---	1:0.85	---	---	1:0.78	---	---
	↓T	6.9; 21.6	---	---	1:1.14	---	---	1:1.06	---	---
2	$\bar{T}$	5.7; 25.0	---	---	1:1.01	---	---	1:0.97	---	---
	↑T	4.5; 28.3	---	---	1:0.91	---	---	1:0.91	---	---
	↓T	100; 23.8	98.2; 22.7	90.4; 23.0	1:1.24	1:1.18	1:1.43	---	1:0.00	1:0.27
3	$\bar{T}$	97.2; 25.4	91.0; 25.4	86.0; 24.8	1:1.38	1:1.26	1:1.32	1:0.71	1:0.12	1:0.26
	↑T	94.4; 27.1	83.9; 26.3	81.6; 26.7	1:1.54	1:1.32	1:1.21	1:0.71	1:0.14	1:0.26
	↓T	91.9; 23.7	88.9; 24.3	59.8; 24.6	1:1.18	1:1.68	1:1.32	1:0.86	1:0.50	1:0.41
4	$\bar{T}$	89.7; 25.5	88.9; 26.1	60.0; 26.4	1:1.22	1:1.19	1:1.45	1:0.48	1:0.55	1:0.53
	↑T	87.6; 27.3	88.9; 28.0	60.3; 28.2	1:1.27	1:0.82	1:1.63	1:0.29	1:0.60	1:0.71
	↓T	100; 22.5	100; 22,3	100; 23.3	1:1.08	1:1.26	1:09	---	---	---
5	$\bar{T}$	98.4; 24.8	96.6; 24.3	91.8; 24.8	1:0.96	1:1.02	1:1.07	1:0.00	1:0.00	1:2.09
	↑T	96.9; 27.2	93.3; 26.4	83.7; 26.4	1:0.86	1:0.85	1:1.28	1:0.00	1:0.00	1:2.09
	↓T	100; 23.4	100; 22.9	100; 23.4	1:1.13	1:1.03	1:1.05	---	---	---
6	$\bar{T}$	100; 25.4	100; 25.0	100; 25.2	1:1.02	1:1.10	1:0.97	---	---	---
	↑T	100; 27.4	100; 27.1	100; 26.9	1:0.93	1:1.18	1:0.91	---	---	---
	↓T	100; 21.7	100; 21.8	100; 22.9	1:1.30	1:1.42	1:1.29	---	---	---
7	$\bar{T}$	100; 24.2	100; 24.2	100; 24.6	1:1.24	1:1.33	1:1.10	---	---	---
	↑T	100; 26.7	100; 26.5	100; 26.4	1:1.17	1:1.27	1:0.97	---	---	---
	↓T	100; 21.0	100; 21.6	100; 22.2	1:1:34	1:1.00	1:1.10	---	---	---
8	$\bar{T}$	100; 23.6	98.7; 24.0	98.0; 23.9	1:1.32	1:1.22	1:1.00	---	1:3.00	1:1.00
	↑T	100; 26.1	97.4; 26.3	96.0; 25.5	1:1.31	1:1.56	1:0.89	---	1:3.00	1:1.00
	↓T	100; 22.2	100; 22.0	100; 22.1	1:1.34	1:1.29	1:1.20	---	---	---
9	$\bar{T}$	100; 24.6	100; 24.0	100; 24.3	1:1:47	1:1.37	1:1.37	---	---	---
	↑T	100; 26.9	100; 25.9	100; 26.4	1:1.59	1:1.44	1:1.56	---	---	---

**Table 7 insects-16-00253-t007:** Results of the multiple linear regression on protective nets’ efficacy ε against SWD (R^2^ = 0.850; Adj. R^2^ = 0.847; F-statistic = 253.8 (*p*-valor = 4.64 × 10^−55^); number of observations = 138).

Variable	Coefficient	Standard Error	t-Value	*p*-Value	95% Conf. Interval
Intercept	226.98	10.273	22.095	<0.001	[206.67, 274.30]
*u* (m/s)	−1.64	0.745	−2.207	0.029	[−3.12, −0.17]
*T* (°C)	−0.97	0.372	−2.605	0.010	[−1.71, −0.23]
*L_px_* (μm)	−0.15	0.005	−26.633	<0.001	[−0.16, −0.14]

**Table 8 insects-16-00253-t008:** Tukey’s HSD test results for air velocity on protective net efficacy.

**Group 1** **(m/s)**	**Group 2** **(m/s)**	**Mean** **Difference**	*p*-Value	95% Conf. Interval	Significant (Reject H0)
0	1.5	18.51	0.002	[6.15, 30.87]	Yes
0	3.0	12.88	0.039	[0.52, 25.24]	Yes
1.5	3.0	−5.63	0.567	[−18.74, 7.48]	No

## Data Availability

The data presented in this study are openly available in FigShare at https://doi.org/10.6084/m9.figshare.27201399 (accessed on 10 October 2024).

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
