# Peer review of "The Efficacy of Protective Nets Against Drosophila suzukii: The Effect of Temperature, Airflow, and Pest Morphology"

_insects, 2025, doi:10.3390/insects16030253_

Round 1

Reviewer 1 Report

Comments and Suggestions for Authors

The paper by Álvarez et al. “Efficacy of Protective Nets against Drosophila suzukii: Abiotic Factors and Influence of Temperature on Body Size” addresses an important issue related to the control of SWD by physical methods designed to reduce, or eliminate, pesticide use.

The research is focused on the study of the effectiveness of crop protection nets, and in particular the authors investigate the structure of the nets by analysing nine commercial types. The parameters considered, which may affect the effectiveness of nets, were mesh size and shape and wind speed, correlated with morphometric analysis of Drosophila adults, sexual dimorphism and the impact of temperature on insect size.

The authors, by setting up a pilot system in the laboratory, show that a high air velocity reduces the effectiveness of the nets; and a loss of effectiveness is also observed at higher temperatures (ranging from 24.1 to 28.2 °C), in the latter case, the temperature, by limiting the size development of the adults makes it easier for them to cross the nets.

The experimental device made by the authors to evaluate the effectiveness of the protection nets was realized by means of a tubular plastic structure divided into individual chambers between which the filter element was placed. And it was equipped with an axial fan and a thermo-anemometer for airflow and temperature control.

Overall, the results confirm that the use of insect nets could be a sustainable alternative to chemical control, provided the choice of net is wisely made. In summary, they suggest that these abiotic (climatic) and morphological factors should be considered before choosing the type of net to use in a specific site.

The paper is correctly written, the English is correct, the experimental logic is clearly described, and the results easily interpreted.

Author Response

Dear Reviewer,

We sincerely appreciate the positive evaluation of our manuscript and the reviewer’s thoughtful summary of our work. We are pleased that the experimental design, results, and writing were well received.

Reviewer 2 Report

Comments and Suggestions for Authors

The authors have tested nine different nets under lab conditions and tested different temperatures, airflow and body morphological parameters to determine the effectiveness of the protective nets against Drosophila suzukii. The authors have done a great job but there is always room for improvement. My comments are:

Title

I would suggest revising the title to accommodate the factors: Influence of Temperature, Airflow, and Pest Morphology on the Effectiveness of Protective Nets against Drosophila suzukii

Abstract: I feel that there is a missing link between the two findings. Try to connect your findings. Moreover, the materials and methods are not specifically mentioned. I recommend adding it. Also, based on your findings, give actionable recommendations of how it could be integrated into the IPM programs.

Introduction

L54: The first line of the paragraph is always a punch line to involve the reader. Make it clear and straightforward.

L54-55: It would prefer to use management instead of control.

L58-59: These are insecticides, not products. Be specific. Also, you can add the word non-target organisms including…

L61-64: I can see redundant content here. Make it a single line which covers everything.

L67: Just use biorational instead of biological and biotechnical. The rest of the sentence will remain the same.

L69-73: Now, here again, you have described a big concept of IPM and then jumped onto the cultural management. Just focus on IPM here.

L74: These practices? Which ones? IPM or Cultural? Be very precise. As I have mentioned before, the first line of paragraph is very important to catch the interest of the reader.

L78-81: Rewrite the sentence for clarity.

L100-102: Your hypothesis statement is much broader than what you have written here. Revise.

Materials and Methods

L118: described in? a previous study or add something to complete this sentence as intext citation style is different for MDPI.

L139: Check for the font.

L171: Manufacturer’s details of entomological aspirator.

L178: Give complete details about the manufacturer according to the style mentioned in author guidelines.

L200: How did you achieve the normal distribution of the data? What transformations have you used?

Results:

You performed ANOVA and found significant single and interactive effects. However, you have not expanded this result to find the differences. For example, let’s take the trait of body length. We do not know that which temperature or which sex has more different from the other. So, make tables and expand these values.

In table 2, the sum of squares is not needed.

Discussion and conclusion seem well written. Excellent.

Author Response

Dear Reviewer,

We sincerely appreciate your valuable comments and suggestions. Below, we provide a point-by-point response addressing each of your observations.

Comments 1: Title - I would suggest revising the title to accommodate the factors: Influence of Temperature, Airflow, and Pest Morphology on the Effectiveness of Protective Nets against Drosophila suzukii

Response 1: We appreciate the reviewer's suggestion and have revised the title accordingly. However, we have slightly adjusted the word order to ensure that readers can immediately identify that the article focuses on protective nets and Drosophila suzukii. Therefore, the revised title will be: "Efficacy of Protective Nets against Drosophila suzukii: the Role of Temperature, Airflow, and Pest Morphology"

----------------------------------

Comments 2: Abstract - I feel that there is a missing link between the two findings. Try to connect your findings. Moreover, the materials and methods are not specifically mentioned. I recommend adding it. Also, based on your findings, give actionable recommendations of how it could be integrated into the IPM programs.

Response 2: We appreciate your comments and agree with your suggestions. We have improved the Abstract to incorporate your recommendations. The final version of the Abstract is as follows:

"Abstract: Drosophila suzukii is an invasive pest that poses a significant threat to fruit crops worldwide, leading to considerable agricultural losses and economic damage. Unlike chemical control measures against D. suzukii, integrating insect-proof nets within an IPM framework offers a more sustainable solution. This study evaluates the efficacy of nine commercial protective nets against this pest, focusing on determining optimal hole dimensions based on the effects of airflow velocity, temperature, and pest morphometry on net performance. To simulate field conditions in the laboratory, we developed a tubular device with three chambers (the tested net placed between two of them), incorporating a fan to generate airflow and a thermo-anemometer. Our results confirm that higher air velocities and elevated temperatures reduce net efficacy. Additionally, morphometric analyses of lab-reared flies revealed significant sexual dimorphism and a strong temperature-size relationship, with flies reared at lower temperatures being consistently larger, an aspect that also affects net effectiveness. These findings highlight the importance of considering both abiotic factors and pest morphology when evaluating protective screens, challenging the assumption that exclusion net efficacy remains constant. Some tested nets proved completely effective against SWD, supporting their use as a preventive measure in IPM programs."

----------------------------------

Introduction

Comments 3: L54: The first line of the paragraph is always a punch line to involve the reader. Make it clear and straightforward.

Response 3: Completely agree. We have changed this first line and the beginning of the paragraph is as follows:
"Despite their environmental and economic drawbacks, insecticides remain the most widely used method for managing this pest [29–31]. Currently, SWD management relies heavily on the application of broad-spectrum insecticides, …"

----------------------------------

Comments 4: L54-55: It would prefer to use management instead of control.

Response 4: We agree. We have changed 'control' to 'management', as you can see in the above response.

----------------------------------

Comments 5: L58-59: These are insecticides, not products. Be specific. Also, you can add the word non-target organisms including…

Response 5: We agree. We have changed these two lines as follows:

"Additionally, these insecticides can have negative effects on non-target organisms, including mammals and beneficial arthropods."
 ----------------------------------

Comments 6: L61-64: I can see redundant content here. Make it a single line which covers everything.

Response 6: We fully agree. We have revised L61-64 into a single sentence as follows:
"The effectiveness of insecticides is increasingly limited due to the emergence of resistance, driven by repeated exposure, short generation time, and high fecundity."

----------------------------------

Comments 7: L67: Just use biorational instead of biological and biotechnical. The rest of the sentence will remain the same.

Response 7: Done.

----------------------------------

Comments 8: L69-73: Now, here again, you have described a big concept of IPM and then jumped onto the cultural management. Just focus on IPM here.

Response 8: We agree with your comment. We have replaced the last sentence of that text ("Cultural management practices include sanitation measures, timely harvested, pruning, irrigation, mulching, and the use of protection nets.") with the following:

"These combined approaches aim to reduce pest populations while minimizing reliance on chemical control."

----------------------------------

Comments 9: L74: These practices? Which ones? IPM or Cultural? Be very precise. As I have mentioned before, the first line of paragraph is very important to catch the interest of the reader.

Response 9: Yes, it is important for the first line to capture the reader's interest. We have rewritten this first line as follows:

"IPM strategies not only effectively manage pests but also meet consumer demand for high-quality fruit with reduced pesticide use."

----------------------------------

Comments 10: L78-81: Rewrite the sentence for clarity.

Response 10: You are right; this sentence is not clear. We have revised it as follows:

"The increasing use of fine mesh netting is driven by the introduction of exotic pests, such as SWD, which are difficult to control with pesticides, as well as increasing social pressure to reduce chemical inputs in agriculture and concerns about the long-term sustainability of chemical-based control methods."

----------------------------------

Comments 11: L100-102: Your hypothesis statement is much broader than what you have written here. Revise.

Response 11: In this case, we respectfully disagree with the comment because our study effectively demonstrates that higher airflow and higher temperatures reduce the effectiveness of the nets. Additionally, we show that net efficacy is not a constant value, as it depends on multiple variables, including pest morphometry. In any case, we have revised the hypothesis for greater clarity and conciseness as follows:

"Our hypothesis is that higher air velocity and temperatures will result in lower efficacy. Additionally, we propose that net efficacy is not a fixed value, as it is influenced by multiple factors, including pest morphometry."

----------------------------------

Materials and methods

Comments 12: L118: described in? a previous study or add something to complete this sentence as intext citation style is different for MDPI.

Response 12: Done.

----------------------------------

Comments 13: L139: Check for the font.

Response 13: Done.

----------------------------------

Comments 14: L171: Manufacturer’s details of entomological aspirator.

Response 14: Done.

----------------------------------

Comments 15: L178: Give complete details about the manufacturer according to the style mentioned in author guidelines.

Response 15: Done.

----------------------------------

Comments 16: L200: How did you achieve the normal distribution of the data? What transformations have you used?

Response 16: The data naturally followed a normal distribution, and no transformations were applied. We conducted a Kolmogorov-Smirnov (K-S) test to determine whether the data follows a normal distribution. The results of the test indicate that the data conforms to a normal distribution.

----------------------------------

Results

Comments 17:You performed ANOVA and found significant single and interactive effects. However, you have not expanded this result to find the differences. For example, let’s take the trait of body length. We do not know that which temperature or which sex has more different from the other. So, make tables and expand these values.

Response 17: Thank you for your suggestion. To address this, we have included post hoc comparisons (Tukey HSD) to clarify which temperature groups and sexes differ significantly in each morphometric trait. These results are now presented in an additional table.

----------------------------------

Comments 18: In table 2, the sum of squares is not needed.

Response 18: In Table 2, the sum of squares is included to provide a detailed breakdown of variance in the two-way ANOVA, helping to interpret the contribution of each factor and their interactions.

----------------------------------

Comments 19: Discussion and conclusion seem well written. Excellent.

Response 19: We appreciate your kind words. Thank you!

Reviewer 3 Report

Comments and Suggestions for Authors

The paper describes a study of the ability of nine different insect netting products with different mesh sizes and characteristics to block the movement of spotted winged drosophila (SWD) adults. The flies seem to be reared at different temperatures for the study. There is a detailed morphometric analysis of adult size and how files of different sizes affect the efficacy of the netting.  There also are detailed measurements of the characteristics of the insect netting.  Previous studies on the efficacy of insect netting to control SWD seem to be present/absences type comparisons with few details about the netting structure.  So, the current study is useful in providing a more detailed examination of the importance of various characteristics of the netting as they influence net efficiency.  One suggested addition is to identify the source (Manufacturer and product number) of the 9 insect nets in the study. The title also could be improved.  The phrase ‘influence of temperature on body size’ is misleading.  That was not the focus of the study. Instead, it was about how body size influenced the efficacy of the protective nets. 

The main purpose of the study is to examine the efficacy of different types of insect netting to prevent the movement of SWD flies.  The morphometric measures of fly size are a useful analysis as it relates to the efficacy of the insect netting.  But the analysis and presentation of results about the fly measurements is more detailed and unnecessarily long for the purpose of the study.  Also Figure 3 is very impressive visually, but I am not clear on how useful or necessary it is for the focus of the paper on net efficacy. The paper would be more useful if the results section and discussion focused more on the net characteristics and efficiency and less on the detailed description of the differences in adult body size. Finding that females are larger than males and flies are larger when reared at lower temperature is nothing new.

Another question or point that needs more clarification is about the rearing conditions and temperature regimes. Line 168 simply states that the temperature varied from 21.0 to 28.5C.  What were the temperature treatments? Were the flies reared at different constant temperatures or in batches under varying temperatures. Also, what were the lab temperature conditions for the net efficacy experiments? Was it a constant ‘room’ temperature or did it vary? This is a key part of the study and the methods for the two temperature interval treatments are not clear at all. Also, how long or how many generations were the flies in colony? Is the range of sizes reported in Table 1 similar to the range of size that would be expected of wild SWD populations? 

In general, the English and grammar are very good. But the text often is somewhat wordy and sometimes repetitive.  This is especially the case for the morphometric analysis results section. The paragraph on lines 384-391 is a key part of the results and should to be expanded. Nets 6 and 7 seem to be completely effective but are not mentioned in this section. The conclusion section also seems long. Can the future work be condensed to one or two sentences? I would also agree with line 600 that a field evaluation of the 9 nets would have been a useful addition to this work.  Maybe this will be the topic of another paper?

One final comment is this paper appropriate for the journal Sustainability?  It seems to be more appropriate for a journal like ‘Insects’ or a pest management journal.  

Author Response

Dear Reviewer,

We sincerely appreciate your valuable comments and suggestions. Below, we provide a point-by-point response addressing each of your observations.

-------------------------------------

Comments 1: One suggested addition is to identify the source (Manufacturer and product number) of the 9 insect nets in the study.

Response 1: We completely agree with your suggestion. However, we approached major insect net distributors to access a comprehensive catalogue from which we could select nets suitable for combating SWD. For this reason, in many cases, the nets are marketed under the distributor’s name, which does not correspond to the original manufacturer or brand. Therefore, we believe it is more appropriate to maintain the codes assigned in the manuscript to avoid confusion.

-------------------------------------

Comments 2: The title also could be improved.  The phrase ‘influence of temperature on body size’ is misleading.  That was not the focus of the study. Instead, it was about how body size influenced the efficacy of the protective nets.

Response 2: We appreciate the reviewer's suggestion and have revised the title accordingly. However, we have slightly adjusted the word order to ensure that readers can immediately identify that the article focuses on protective nets and Drosophila suzukii. Therefore, the revised title will be: "Efficacy of Protective Nets against Drosophila suzukii: the Role of Temperature, Airflow, and Pest Morphology"

-------------------------------------

Comments 3: But the analysis and presentation of results about the fly measurements is more detailed and unnecessarily long for the purpose of the study.  Also Figure 3 is very impressive visually, but I am not clear on how useful or necessary it is for the focus of the paper on net efficacy.

Response 3: Thank you for your comment. We believe that Figure 3 is important because it visually highlights the size differences driven by temperature and sex. Moreover, it conveys a key message for the design and selection of the most suitable nets: rather than relying on average body size, lower percentiles of the population size distribution should be considered, since using mean values could allow the passage of smaller individuals (i.e., all individuals with a size below the average). For this reason, we consider it valuable to retain this figure.

-------------------------------------

Comments 4: The paper would be more useful if the results section and discussion focused more on the net characteristics and efficiency and less on the detailed description of the differences in adult body size. Finding that females are larger than males and flies are larger when reared at lower temperature is nothing new.

Response 4: Thank you for pointing this out. We have now added new observations in section 3.3. Efficacy of protective nets in excluding D. suzukii.

-------------------------------------

Comments 5: Another question or point that needs more clarification is about the rearing conditions and temperature regimes. Line 168 simply states that the temperature varied from 21.0 to 28.5C.  What were the temperature treatments? Were the flies reared at different constant temperatures or in batches under varying temperatures. Also, what were the lab temperature conditions for the net efficacy experiments? Was it a constant ‘room’ temperature or did it vary? This is a key part of the study and the methods for the two temperature interval treatments are not clear at all.

Response 5: Thank you for your comment. We appreciate the opportunity to clarify these aspects of our study. The flies were reared in a non-air-conditioned laboratory, where temperature was influenced by outdoor conditions. The rearing period extended from October 2022 to July 2023, leading to natural temperature variations between 21.0 and 28.5 °C. Therefore, the flies were not reared under controlled constant temperatures, but rather under fluctuating environmental conditions.

Regarding the net efficacy experiments, these were conducted under laboratory conditions at ambient temperature, the same conditions under which the rearing of the flies was conducted.

-------------------------------------

Comments 6: Also, how long or how many generations were the flies in colony? Is the range of sizes reported in Table 1 similar to the range of size that would be expected of wild SWD populations?

Response 6: We did not monitor the number of generations in the colony, but the flies were maintained in culture for a period of 10 months, which coincided with the duration of the experimental phase of this study. Regarding the second point, it is possible that the body size of wild SWD populations exhibits some variation due to factors such as diet quality, resource availability, and other environmental conditions. While our measurements provide a representative range for the individuals reared under laboratory conditions, it is possible that these may not fully capture the potential variability found in wild populations.

-------------------------------------

Comments 7: But the text often is somewhat wordy and sometimes repetitive.  This is especially the case for the morphometric analysis results section.

Response 7: Thank you for your valuable feedback. We have carefully reviewed the text, particularly the section on the morphometric analysis results, to improve clarity and reduce repetition. We have restructured some sentences and eliminated redundant phrases to ensure a more concise presentation of the results without sacrificing any essential details. We hope these revisions address your concerns and enhance the overall readability of the manuscript.

-------------------------------------

Comments 8: The paragraph on lines 384-391 is a key part of the results and should to be expanded. Nets 6 and 7 seem to be completely effective but are not mentioned in this section.

Response 8: Thank you for your comment. We have expanded the paragraph on lines 384-391 to provide more detailed information. Additionally, we have added two more paragraphs to further clarify the results. In these revisions, we have explicitly highlighted that nets 6 and 7 are completely effective, as you pointed out. We believe these changes better address the significance of these findings in the context of the overall study.

-------------------------------------

Comments 9: The conclusion section also seems long. Can the future work be condensed to one or two sentences?

Response 9: Done.

-------------------------------------

Comments 10: I would also agree with line 600 that a field evaluation of the 9 nets would have been a useful addition to this work.  Maybe this will be the topic of another paper?

Response 10: Thank you for your suggestion. We agree that such an evaluation would be a valuable addition to this study. However, as our research team is relatively small, we currently do not have the capacity to conduct fieldwork. Additionally, SWD is not present in our local area, which limits our ability to carry out such studies. Nevertheless, we hope that this work could be pursued in the future, potentially in collaboration with other teams or institutions with access to affected areas.

-------------------------------------

Comments 11: One final comment is this paper appropriate for the journal Sustainability?  It seems to be more appropriate for a journal like ‘Insects’ or a pest management journal.

Response 11: Thank you for your valuable comment. You are absolutely right that this paper may be more suited for a journal like Insects or one focused on pest management. In fact, the article is currently under consideration by the editorial team of Insects, and we believe it aligns better with the scope and audience of that journal.

Round 2

Reviewer 2 Report

Comments and Suggestions for Authors

The authors have addressed most of the comments.

Author Response

Dear Reviewer,

We sincerely appreciate the positive evaluation of our manuscript. Your constructive feedback and encouraging comments are invaluable to us.

Thank you for your time and effort in reviewing our work,

Best regards,

The Authors

Reviewer 3 Report

Comments and Suggestions for Authors

The revision of Insects-3279579 Alveraz et al. is acceptable.  The authors have addressed all of my comments satisfactorily.  I particularly like the addition text on line 448-472 about specific net efficiency.  I see figure 3 is still there which is acceptable. Also, the authors have added 2 more tables, but the additional tables provide useful statistical information.  I notice that the authors did not provide Manufacturer and product number information for the 9 net types as I suggested.  Is this information available elsewhere such as in reference 66 or 69?  This information would seem to be useful for other entomologist/agriculturists to implement the use of netting for control of SWD. 

Author Response

Dear Reviewer,

We greatly appreciate your insightful comment and recommendations. Below, we have provided a detailed response to address your observations.

Comments 1: I notice that the authors did not provide Manufacturer and product number information for the 9 net types as I suggested.  Is this information available elsewhere such as in reference 66 or 69?  This information would seem to be useful for other entomologist/agriculturists to implement the use of netting for control of SWD. 

Response 1: We understand the reviewer’s concern and apologize for not being able to provide the manufacturer and product information for all the nets. The reason is that we do not have this information for every net used in our study. Over the past two decades, we have assembled a large collection of nets (hundreds) from both manufacturers and distributors. In the case of nets obtained from distributors, some were rebranded and marketed under the distributor’s name, making it impossible to identify the original manufacturer.

This is precisely the situation with some of the nets in this study. We only know the commercial names and origins of some of them, and we believe it would be inconsistent to provide that information selectively. However, we consider that the most valuable information for entomologist/agriculturists is the detailed geometric characterization of each net, which is fully provided in Table 5 (formerly Table 3). With these specifications (together with insect morphology and the efficacy data provided in Table 6 (formerly Table 4)), any interested party has all the necessary data to select an appropriate commercial net or enable any manufacturer to produce an effective net against SWD based on the comprehensive information provided in the manuscript.

We would also like to point out that, in our experience, articles that do provide commercial names often lead to manufacturer websites where geometric details of the nets are scarce; in most cases, only the 'mesh' (threads per inch in the warp direction) is provided, which is insufficient. In contrast, our study offers a comprehensive geometric description, which we believe is more practical and informative.

In summary, we would be happy to include the commercial details if we had them for all nets, but we do not. We believe that providing partial information could be misleading, whereas the geometric data we present is complete and more valuable for both researchers and practitioners.

The requested information is not available in references 66 or 69, for the same reasons previously mentioned. Moreover, both references focus on different groups of nets, which are not the same as those analysed in our study. However, as in our manuscript, these references also offer a detailed geometric analysis of the nets.

Thank you for the time and effort in reviewing our work.

Best regards,

The Authors